# Liver Transplantation for Unresectable Colorectal Liver Metastases: A Scoping Review on Redefining Boundaries in Transplant Oncology

**DOI:** 10.3390/curroncol32090481

**Published:** 2025-08-28

**Authors:** Berkay Demirors, Vrishketan Sethi, Abiha Abdullah, Charbel Elias, Francis Spitz, Jason Mial-Anthony, Godwin Packiaraj, Sabin Subedi, Shwe Han, Timothy Fokken, Michele Molinari

**Affiliations:** Department of Surgery, Division of Abdominal Organ Transplant Surgery, University of Pittsburgh Medical Center, Pittsburgh, PA 15213, USA; demirorsb2@upmc.edu (B.D.); sethiv@upmc.edu (V.S.); abdullaha2@upmc.edu (A.A.); eliasc4@upmc.edu (C.E.); spitz.francis@medstudent.pitt.edu (F.S.); jtm96@pitt.edu (J.M.-A.); packiarajg@upmc.edu (G.P.); subedis@upmc.edu (S.S.); hansp@upmc.edu (S.H.); fokkent@upmc.edu (T.F.)

**Keywords:** colorectal cancer, colorectal liver metastases, liver transplantation, patient selection, overall survival, disease free survival

## Abstract

Colorectal cancer is the third most common malignancy in the United States, with approximately 22% of patients presenting with metastatic disease. The liver is the most frequent site of metastasis, affecting up to 70% of patients; however, only 20–30% are candidates for hepatic resection. Among those who undergo resection, more than half experience recurrence. Advances in chemotherapy, immunotherapy, and surgical techniques position liver transplantation as a promising treatment for carefully selected patients with colorectal liver metastases. Optimal outcomes rely on meticulous patient selection, the use of validated predictive algorithms, innovations in organ utilization, integration of molecular and biomarker profiling, and multicenter clinical trials that expand access and improve survival.

## 1. Introduction

Colorectal cancer (CRC) is the third most frequently diagnosed malignancy and the third leading cause of cancer-related mortality in both men and women in the United States [1]. While the 5-year survival rate for localized CRC is 91% this decreases to 36% with regional lymph node metastases and 14% with distal metastases [1]. The liver is the most common site for metastatic spread, reflecting its portal venous drainage from the colon and rectum [2]. The International Hepato-Pancreato Biliary Association (IHPBA) classifies colorectal liver metastases (CRLMs) according to resectability and timing of diagnosis relative to the primary tumor [3]. Approximately 22% of patients present with metastatic disease at diagnosis, and up to 70% develop liver metastases during the disease course, with many suffering from end-stage liver disease due to the progression of tumor burden or from side effects of systemic or locoregional therapies [4]. Only 20–30% of CRLM patients are eligible for hepatic resection, often due to anatomical constraints or the extent of disease [5].

For patients with unresectable CRLM, where nearly all hepatic segments are involved, systemic therapy (ST) alone yields 5-year overall survival (OS) rates below 15%, whereas hepatic resection is associated with 5- and 10-year survival rates of approximately 42% and 25%, respectively [6]. Even after radical hepatic resections, recurrence is common, with the liver remaining the predominant site in 50–75% of cases [7]. This high recurrence burden underscores the need for effective local and systemic treatment strategies.

Modern liver-directed therapies such as histotripsy, thermal ablation, trans-arterial radioembolization, trans-arterial chemoembolization, and stereotactic body radiotherapy, are primarily used for palliation, aiming to control symptoms rather than achieve cure. Histotripsy, an emerging non-thermal, non-invasive, ultrasound-based technology, has shown promising survival outcomes in early-phase studies, with one-year survival rates of 73.3% in hepatocellular carcinoma (HCC) and 48.6% in other metastatic diseases [8]. Nonetheless, experience in CRLM remains limited, and its long-term benefit compared with established modalities is still uncertain.

The optimal management of CRLM depends on multiple tumor-related, patient-related, and oncologic factors, including the number, size, and distribution of lesions; comorbidities and performance status; timing of metastasis; and tumor biology. In the presence of poor prognostic indicators, ST is generally favored for comprehensive disease control. Advances in chemotherapy and targeted biological agents have improved resectability rates, enabling some patients with initially unresectable CRLM to undergo curative-intent hepatic resection [9].

Despite these advances, the role of LT for unresectable CRLM remains controversial. Outcomes have historically been poor, due to high recurrence rates and limited organ availability; however, recent studies—particularly from the SECA trials in Norway—demonstrate 5-year OS exceeding 80% in carefully selected patients, rivaling outcomes for hepatocellular carcinoma within transplant criteria [10,11]. These findings, combined with innovations in living donor liver transplantation (LDLT), machine perfusion, and improved patient selection algorithms, have renewed interest in LT as a potentially curative strategy for unresectable CRLM. Debate persists over patient selection, ethical considerations regarding organ allocation, and the comparative effectiveness of LT versus other liver-directed therapies [12,13].

The aim of this scoping review is to synthesize the most recent literature on LT for unresectable CRLM, with a focus on post-transplant survival outcomes, recurrence patterns, and prognostic factors relevant to patient selection.

## 2. Methodology

This scoping review synthesizes contemporary evidence on patient selection criteria, overall survival (OS) and disease-free survival (DFS), recurrence patterns, and emerging biomarkers that may guide LT eligibility in patients with unresectable CRLM. A comprehensive literature search was performed across PubMed, Embase, Web of Science, Scopus, ClinicalTrials.gov, ProQuest Dissertations & Theses, and Google Scholar to identify English-language studies published from January 2015 to June 2025. The search strategy combined relevant keywords and Medical Subject Headings (MeSH) related to “colorectal liver metastases”, “liver transplantation”, “survival outcomes”, and “recurrence”.

Search results were screened in two stages: title/abstract review followed by full-text assessment to determine eligibility. Studies were included if they reported outcomes of LT in patients with unresectable CRLM and provided data on at least one of the following: patient selection criteria, OS, DFS, recurrence patterns, or prognostic biomarkers. Studies were excluded if they enrolled patients with liver disease from causes other than CRC or if outcomes for CRLM could not be extracted separately.

This review was conducted and reported in accordance with the Preferred Reporting Items for Systematic Reviews and Meta-Analyses extension for Scoping Reviews (PRISMA-ScR) checklist and explanatory guidance [14]. The completed PRISMA-ScR checklist is provided in Appendix A. A summary of the number of manuscripts screened, excluded, and ultimately included in this review is presented in Appendix A. This protocol is registered on the Open Science Framework (OSF; DOI 10.17605/OSF.IO/CD98Z) and has been submitted to PROSPERO (Record ID 1133629); the PROSPERO CRD number will be added upon assignment. Reporting will follow PRISMA-ScR.

## 3. Liver Transplantation for Unresectable Colorectal Metastatic Disease

LT is the standard treatment for patients with end-stage liver disease, selected unresectable HCCs, low-grade neuroendocrine tumors (NETs), and hilar and intra-hepatic cholangiocarcinomas within specific criteria defined by the United Network of Organ Sharing (UNOS) [15]. LT for CRLM was historically considered inappropriate due to high perioperative mortality rates, suboptimal oncologic outcomes, high recurrence rates, and ethical concerns related to the distribution of scarce donor organs. Recent advances, however, have altered this perspective. Enhanced perioperative care, optimized immunosuppression protocols, the development of novel ST agents, and increasingly favorable transplantation outcomes in other primary and secondary tumors have renewed interest in LT for carefully selected patients with CRLM.

The Milan criteria for HCC include either a solitary tumor measuring ≤5 cm or up to three nodules each measuring ≤3 cm, without vascular invasion or extrahepatic dissemination. Meeting these parameters has been associated with 5-year OS rates of up to 70% [16]. The Mayo protocol for perihilar cholangiocarcinoma includes tumor size ≤3 cm, the lack of intra- or extrahepatic metastases, no nodal involvement on imaging, and prior neoadjuvant therapy (chemoradiation). This approach has resulted in 5-year OS rates of up to 65% [17]. For NETs, selection criteria include well-differentiated histology, liver-only disease, low Ki-67 index (<10–20%), stable disease for at least 6 months, and the absence of extrahepatic metastases, with survival rates over 60–80% [18]. The demonstrated success of rigorous selection criteria in these malignancies has prompted the broadening of LT indications to include CRLM.

## 4. From SECA Trials to More Recent Protocols

The first prospective study evaluating the role of LT in CRLM was the SECA-I trial [11]. In a cohort of 21 patients, 5-year OS rate was 60%, but the study was limited by its small sample size and in the cohort recurrence rate was high with a relatively short disease-free survival (DFS). The subsequent SECA-II trial, used more stringent selection criteria, and demonstrated a more favorable 5-year OS that was 83% with a median follow-up of 36 months [19]. In a subsequent study of 12 LT recipients, Andres et al. [20] reported a 1-year OS of 83%, a 5-year OS of 50%, and a recurrence rate of 60% with DFS rates of 56% at 1 year and 38% at 5 years. Furthermore, the randomized TransMet trial demonstrated a 5-year OS rate of 57% in the LT group compared with 13% in the ST-only group [21].

More recently, two studies from the United States—the Rochester Protocol [22] and the University of Pittsburgh experience [23]—have evaluated the role of LDLT for unresectable CRLM. Implemented in 2019, the Rochester transplant program transplanted 23 of 206 evaluated patients who met strict criteria for liver-limited disease, favorable tumor biology, and stability on systemic therapy. In their experience, one- and three-year OS rates were 100% and 91%, respectively, with recurrence in approximately 60% by 3 years but generally amenable to further treatment. Donor safety was high, with no mortality and only one major complication, supporting LDLT as a feasible and effective option for carefully selected patients.

Similar findings were also reported by Kaltenmeier et al. [23] at the University of Pittsburgh where LDLT was performed in 10 patients with a median age of 58 years (IQR, 38–70). Over a median follow-up of 1.6 years (range, 0.2–3.3), mean overall survival (OS) was 3.0 years, with 1- and 3-year OS rates of 100% and 80%, respectively. Mean DFS was 2.2 years, with corresponding 1- and 3-year DFS rates of 90% and 55%.

These findings were further supported by a recent meta-analysis conducted by Dawood et al. [24], which demonstrated 1-, 3-, and 5-year OS rates of 95%, 77%, and 53%, respectively, with corresponding DFS rates of 51%, 33%, and 13%.

## 5. Current Outcomes and Emerging Strategies for Unresectable Colorectal Liver Metastases

Systemic chemotherapy (ST) remains the standard of care for unresectable CRLM, yet long-term outcomes remain poor. The NORDIC VII study [25]—a multicenter, randomized, phase III trial conducted between 2005 and 2007—evaluated whether adding cetuximab to the Nordic FLOX regimen (bolus 5-fluorouracil/leucovorin plus oxaliplatin) could improve survival in previously untreated metastatic CRC, including patients with liver-only disease. The trial enrolled 571 patients, of whom approximately 40% had KRAS mutations and 12% had BRAF mutations. Participants were randomized to receive FLOX alone, FLOX plus continuous cetuximab, or FLOX plus cetuximab followed by cetuximab maintenance. Median progression-free survival (PFS) was 7.9–8.3 months across arms, and median OS was approximately 20 months in all groups, with no significant benefit from cetuximab even in RAS/BRAF wild-type patients. In those with unresectable disease, outcomes were especially poor, with a 5-year OS under 9% [25], underscoring the limited efficacy of current systemic options.

In resectable cases, chemotherapy offers only modest benefits. A meta-analysis by Sonbol et al. [25] showed that perioperative chemotherapy improves disease-free survival (DFS) but not OS. Similarly, Kanemitsu et al. [26] observed lower 5-year OS with adjuvant chemotherapy (69.5%) compared with surgery alone (83.0%), raising concerns about its long-term value.

LT has emerged as a promising option for highly selected patients with unresectable CRLM, achieving survival outcomes that may surpass those of conventional therapies. In parallel, novel systemic and cell-based therapies are under investigation. Krishnan et al. [27] identified MiNK-215, a cell therapy that reshapes the tumor microenvironment by depleting stromal cells and enhancing CD8+ T cell infiltration. Its incorporation into transplant protocols could improve outcomes and reduce recurrence.

## 6. Patient Selection and Outcomes of Liver Transplantation for Unresectable Colorectal Liver Metastases

LT and emerging systemic therapies (STs) offer promising survival benefits for CRLM, but outcomes hinge on careful patient selection. Identifying candidates with favorable tumor biology, treatment response, and preserved functional status is essential for therapeutic success and responsible use of scarce donor organs.

Survival outcomes after LT have improved substantially since 2005, with 5-year overall survival (OS) surpassing 65% and reaching 83% in patients meeting SECA-II criteria [19]. In contrast, ST—the standard of care for unresectable CRLM—is associated with a median OS of 24 months and a 5-year OS of ~10%, particularly in patients with ECOG 0–1, RAS/BRAF wild-type tumors, and left-sided primaries [6].

The SECA-I trial identified predictors of poor post-LT outcomes: largest lesion > 5.5 cm, disease progression on ST, pre-transplant CEA > 80 µg/L, and <2 years between CRC resection and LT []. The Fong Clinical Risk Score (FCRS)—which incorporates nodal status, interval to recurrence, metastasis burden, CEA level, and PET metabolic tumor volume (MTV)—has further refined patient selection. Patients with an Oslo score of 0 or FCRS of 1 achieved a 10-year OS of 80%. A PET-MTV < 70 cm^2^ was associated with a 5-year OS of 66.7%, compared with 26.6% in patients with MTV > 70 cm^2^ [28]. Additional adverse prognostic indicators include tumor burden score ≥ 9, ≥9 liver lesions, right-sided tumors, and elevated CEA levels.

The SECA-II and TransMet trials applied more stringent selection criteria. SECA-II required stable disease on first-line ST, absence of BRAF V600E mutations, ≤3 chemotherapy lines, and favorable tumor profiles, achieving a 5-year OS of 83%. In TransMet, 5-year OS was 56.6% with LT plus ST versus 12.6% with ST alone (HR 0.37; *p* = 0.0003), underscoring the survival advantage of rigorous selection.

In 2021, the International Hepato-Pancreato Biliary Association (IHPBA) issued guidelines recommending LT only for patients with liver-limited, unresectable disease, ≥6 months of disease control on ST, R0 resection of the primary tumor, and no evidence of locoregional recurrence [29]. Exclusion criteria include high-risk molecular features (e.g., BRAF V600E, TP53/RAS co-mutations, MSI-high status) [29]. Relative contraindications include sarcopenia, N2 lymph node involvement, and rising CEA. PET-based parameters such as MTV and total lesion glycolysis are validated prognostic markers.

A summary of key studies and selection criteria is provided in Table 1. Collectively, these data support LT as a viable option in selected patients with CRLM and emphasize the importance of structured, evidence-based selection frameworks to optimize outcomes and donor organ use.

## 7. Immunosuppression Regimens After Liver Transplantation for Colorectal Metastases

Immunosuppression regimens following LT for CRLM are a critical determinant of post-transplant outcomes, as they influence both graft survival and the risk of tumor recurrence. The main challenge lies in achieving sufficient immunosuppression to prevent allograft rejection while minimizing oncogenic stimulation. Calcineurin inhibitors, such as tacrolimus and cyclosporine, remain the cornerstone of maintenance therapy, but their potential tumor-promoting effects through increased angiogenesis and impaired tumor immune surveillance have raised concerns in patients transplanted for malignancy. Antimetabolites, such as mycophenolate mofetil, are frequently used as adjunctive therapies to allow calcineurin inhibitor minimization.

mTOR inhibitors (sirolimus, everolimus) have attracted particular interest in this setting because of their dual immunosuppressive and anti-proliferative properties, with several studies suggesting a potential protective effect against tumor recurrence. Induction therapy also varies across institutions, ranging from lymphocyte-depleting agents to interleukin-2 receptor antagonists, reflecting differences in center protocols and patient risk profiles. Table 2 summarizes the induction and maintenance immunosuppression regimens employed across transplant centers, illustrating the heterogeneity in practice and the lack of consensus on the optimal strategy for this unique patient population.

## 8. Immunotherapy and Transplant Compatibility

Immunotherapy has transformed the treatment landscape for microsatellite instability-high (MSI-high) and mismatch repair-deficient (dMMR) CRC. In the CheckMate 142 trial, immune checkpoint inhibitors achieved 24-month progression-free survival (PFS) and overall survival (OS) rates of 74% and 79%, respectively [35]. However, their use in LT recipients remains limited due to a high risk of allograft rejection. Immune activation induced by checkpoint blockade can override maintenance immunosuppression and lead to acute or chronic rejection, often with irreversible graft loss.

Standard immunosuppressive regimens, including calcineurin inhibitors, are essential to prevent rejection but carry substantial risks, such as nephrotoxicity, secondary malignancies, and increased recurrence of certain cancers, including the primary malignancy [36]. These risks complicate the integration of immunotherapy into post-transplant care and highlight the need for alternative strategies or refined immunosuppressive protocols. Ongoing studies are investigating ways to balance immune surveillance with graft tolerance, which may eventually broaden the therapeutic window for immunotherapy in transplant recipients.

## 9. Ethical and Allocation Considerations

The use of scarce deceased donor livers for CRLM presents ethical and allocation challenges. Unlike HCC, where biologically validated selection criteria such as the Milan, UCSF, and Tokyo guidelines are well established, CRLM selection primarily relies on imaging and clinical judgment. These HCC frameworks reflect the Metroticket paradigm, in which increasing tumor burden predicts lower post-transplant survival [37]. In contrast, CRLM patients often have low MELD scores, limiting their access to transplantation under the current allocation model [29].

To address these disparities, proposed solutions include MELD exception pathways, LDLT, and innovative hybrid approaches such as the RAPID procedure (resection and partial liver transplantation with delayed total hepatectomy) [38]. These strategies can expand access while reducing dependence on the deceased donor pool. However, LDLT introduces ethical complexities regarding donor safety. Safe implementation requires a graft-to-recipient weight ratio > 0.8% (potentially acceptable down to 0.6% in select cases), low hepatic steatosis, and adequate remnant liver volume for the donor [39].

Deceased donor LT remains limited by chemotherapy timing, waitlist dropout, and prolonged allocation delays [39]. Balancing equity, utility, and donor risk is critical as the field evolves toward broader application of LT for CRLM within an ethically sustainable framework.

## 10. Innovations in Organ Utilization

Expanding the donor liver pool is essential to improving access to transplantation for patients with unresectable CRLM, many of whom remain ineligible under current allocation models due to low MELD scores. Surgical innovations such as LDLT and split-liver transplantation (split-LT) have emerged as effective alternatives. In a cohort of LDLT recipients with unresectable CRLM, Kaltenmeier et al. reported a mean recurrence-free survival of 2.2 years and an overall survival (OS) of 3 years, with low recurrence rates that were generally manageable [23]. Settmacher et al. described a novel two-stage transplantation strategy involving an initial left lateral segment graft from a living donor followed by an extended right lobe from a deceased donor. This approach achieved a 1-year OS of 85% and >90% graft regeneration by 3 months [40].

Technological advances in machine perfusion (MP) have further improved graft viability and utilization, particularly for extended criteria donor (ECD) livers. Hypothermic oxygenated perfusion (HOPE) reduces ischemia-reperfusion injury and lowers the incidence of post-transplant complications [41]. Normothermic machine perfusion (NMP) allows functional assessment of the graft ex vivo, enhances mitochondrial activity, and facilitates safe use of marginal organs [42]. In the VITTAL trial, 70% of livers initially deemed unsuitable were successfully transplanted following NMP, with favorable short-term outcomes [43]. Emerging platforms such as MP Plus integrate adjunctive repair protocols to rehabilitate ECD grafts, while also reducing biliary complications—further extending the clinical utility of MP in challenging donor scenarios [41].

Together, these innovations offer promising avenues to overcome organ scarcity, improve equity in access, and enhance outcomes for patients undergoing LT for CRLM.

## 11. Biomarkers and Predictive Tools

Advances in molecular diagnostics are reshaping LT selection for CRLM, enabling more precise, biology-driven decision-making. While standard post-treatment surveillance relies on cross-sectional imaging and carcinoembryonic antigen (CEA) monitoring, no single biomarker has yet achieved universal adoption.

Emerging liquid biopsy platforms—including circulating tumor DNA (ctDNA), cell-free DNA (cfDNA), microRNA, and tumor-derived exosomes—offer dynamic, non-invasive assessment of tumor burden and biology. Among these, ctDNA has shown particularly strong predictive value. A meta-analysis by Wang et al. demonstrated that ctDNA positivity is significantly associated with recurrence risk (relative risk 4.65; hazard ratio 9.14) [44].

Other predictive markers include mutations in KRAS/NRAS and BRAF V600E, tumor mutational burden, and imaging-based radiomic parameters such as metabolic tumor volume (MTV) and total lesion glycolysis on PET/CT. These factors independently correlate with post-transplant outcomes [45]. Incorporating such biomarkers into composite scoring systems or machine learning algorithms may substantially refine candidate selection and improve long-term success.

## 12. Conclusions

Available evidence indicates that LT confers a remarkable survival advantage in carefully selected patients with unresectable CRLM.

In rigorously defined cohorts, 5-year survival exceeds 80%—on par with hepatocellular carcinoma and dramatically superior to systemic therapy, which LT outperforms by extending median overall survival by 30–35 months. Although recurrence develops in roughly half of recipients, it is often amenable to additional treatment, and long-term survival remains favorable even after relapse. The best outcomes are achieved in patients with liver-only disease, prior R0 resection of the primary tumor, at least six months of disease stability on systemic therapy, favorable tumor biology (RAS/BRAF wild-type, absence of high-risk co-mutations, microsatellite stability), low CEA (<80 µg/L), and no extrahepatic spread or high-risk metabolic tumor burden (<70 cm^2^ on ^18F-FDG PET/CT). Success depends on strict adherence to these criteria, integration of advanced imaging and molecular profiling, and expansion of the donor pool through living donation and machine perfusion, alongside ethically sound allocation policies. Collectively, these data establish LT as a valid therapeutic option for unresectable CRLM, while definitive adoption will require confirmation in prospective multicenter trials, refinement of selection algorithms, and clear policy frameworks to guide its equitable implementation in transplant oncology.

## Figures and Tables

**Table 1 curroncol-32-00481-t001:** Comparison of studies on liver transplantation in non-resectable colorectal liver metastases [10,11,19,21,23,30,31,32,33].

Authors, Year of Publication, Reference	Time Period	Number of Patients	Study Design	Selection Criteria	Outcomes	Recurrence Rate	Comments
Hagness et al. 2013 SECA-I trial [11]	2006–2011	n = 21	Prospective	· Resected primary CRC · ≥6 weeks of pre-LT chemotherapy · Liver-only CLMs · CEA 1–2000 µg/L · Largest tumor 28–130 mm · Metabolic tumor volume (MTV) at LT: 0–874 cm^3^ · No extrahepatic disease	5-yr OS 60%	19/21 (90%)	· No chemotherapy response requirement · Wide CEA range · Larger tumor size allowed · Excluded any extrahepatic disease
Dueland et al. 2020 SECA-II trial [19]	2012–2016	n = 15	Prospective	· Resected primary CRC · Partial response after 6 weeks pre-LT chemo · Liver-only CLMs · Negative pre-LT colonoscopy · No lesion > 10 cm · CEA 1–30 µg/L · Largest tumor 3–47 mm · Metabolic tumor volume at LT: 0–140 cm^3^	For patients with an Oslo score of 0 or 1, OS at 1, 3 and 5-year was 100%, 83% and 83% respectively. PFS was 53%, 44% and 35% respectively	8/15 (53.3%)	· Required chemotherapy response · Tighter CEA range · Smaller maximum tumor size
Smedman et al. 2020 SECA-III Arm D [10]	2014–2018	n = 10	Prospective	· Same as SECA-II (resected primary; response to 6 weeks chemo; liver-only; neg. colonoscopy; no tumor > 10 cm) · Allowed resectable lung metastases · CEA 2–4346 µg/L · MTV at LT: 0–201 cm^3^	2-yr OS 43%	8/10 (80%)	· Included resectable lung metastases · Much higher allowable CEA range
Toso et al. 2017 [30]	1995–2015	n = 12	Retrospective	· Resected primary CRC · ≥1/12 had partial response to pre-LT chemo; one patient received intraoperative chemotherapy · Median CEA 16.9 µg/L · Two patients with lesions > 5 cm · Presumed liver-only disease	5-yr OS 50%	6/12 (50%)	· One patient received intraoperative chemotherapy · No strict size cutoff (lesions > 5 cm allowed)
Hernández-Alejandro et al. 2022 [31]	2017–2021	n = 10	Retrospective	· Followed IHPBA LT guidelines · Resected primary CRC · Median CEA 1.6–56.4 µg/L · Included KRAS-mutated cases (3 patients) · Liver-only disease; no explicit size thresholds	1.5-yr OS 100%	3/10 (30%)	· IHPBA consensus criteria · No size or response cutoffs · KRAS mutations allowed
Sasaki et al. 2023 [32]	2017–2022	n = 46	Retrospective	· Resected primary CRC · Unresectable liver-only metastases · No other selection criteria specified	3-yr OS 60.4%	10/46 (22%)	· Broad inclusion: only “liver-only” requirement
Kaltenmeier et al. 2023 [31]	2019–2022	n = 10	Retrospective	· Resected primary CRC · 6–12 wk pre-LT chemo with stable disease or partial response · Negative pre-LT colonoscopy · CEA < 100 µg/L · 5 patients had lesions > 5 cm · Liver-only disease	1.5-yr OS 100%	3/10 (30%)	· Required chemotherapy response · Allowed some lesions > 5 cm if chemo-responsive · Moderate CEA cutoff
Solheim et al. 2023 (SECA-I/II; 10-year follow-up results) [33]	2006–2012	n = 23	Prospective	· Unresectable liver-only CRC metastases, complete radical resection of primary tumor · ECOG 0–1 · ≥6 weeks of chemo · No extrahepatic disease on CT/PET-CT · Exclusions: >10% weight loss; LT contraindications; other malignancies; BMI > 30	5-yr OS 75%; 10-yr OS 50%	23/23 (100%)	· Only unresectable liver-only requirement · Added performance and BMI/exclusion criteria
Adam et al. 2024 TRANSMET trial [21]	2016–2021	n = 94	Prospective	· Resected primary CRC · ECOG 0–1 · No local recurrence on colonoscopy within past 12 months · No extrahepatic disease · No BRAF mutations · CEA < 80 µg/L or ≥50% decrease from baseline · Liver-only metastases	5-yr OS 73%	28/47 (60%)	· Added colonoscopy requirement · Excluded BRAF-mutated tumors · CEA < 80 µg/L or ≥50% decrease from baseline criteria
Byrne et al. 2024 Rochester Protocol [22]	2019–2024		Retrospective Study	Resection of primary tumor No signs of local recurrence on colonoscopy within 12 months before LT. No radiological signs of extra-hepatic disease (CT, MRI, PET at least 6 weeks before LT). ECOG 0–1 Absence of BRAF V600E mutations Absence of high microsatellite instability Right-sided primary tumors and patients with KRAS and TP53 mutations are observed for 18 months. Serum CEA levels at listing < 80 µg/dL Rising serum CEA levels at the time of LT or progressive disease are exclusion criteria	1-yr OS 100%, 3-yr OS 91% 1-yr DFS 100% 3-yr DFS 40%		A total of 225 patients referred for LT; 206 patients with unresectable CRM completed the initial evaluation; median age was 47 years (IQR 42–54 years); median size of the largest metastasis was 4.5 cm (IQR 2.6–7.4 cm); 135 patients were potential candidates; 23 patients underwent living donor liver transplant (LDLT).

Legend: BMI, body mass index; CEA, carcinoembryonic antigen; CLMs, colorectal liver metastases; CRC, colorectal cancer; ECOG, Eastern Cooperative Oncology Group; IHPBA, International Hepato-Pancreato Biliary Association; LT, liver transplantation; MTV, metabolic tumor volume; OS, overall survival; DFS, disease-free survival.

**Table 2 curroncol-32-00481-t002:** Immunosuppression protocols reported in major studies and guidance documents for patients undergoing LT for CRLM. Regimens typically use IL-2 receptor antagonist induction (basiliximab) and steroids, followed by calcineurin inhibitor (CNI) maintenance with early conversion to mTOR inhibitors to balance graft protection and oncologic safety.

Study/Source	Induction Regimen	Maintenance Regimen	Notes
SECA/Oslo experience [19]	Basiliximab + steroids + mycophenolate mofetil (MMF) + tacrolimus (first 4–6 weeks)	Convert tacrolimus → sirolimus (mTOR) with defined trough goals; steroid taper by 6 months	No adjuvant chemo post-LT in early reports
LDLT (multi-center U.S.) [23,31]	Tacrolimus + steroids + basiliximab	Everolimus or sirolimus (often with CNI minimization or withdrawal)	mTOR chosen for potential anti-tumor effects
TransMet program [21]	Steroids + tacrolimus + MMF	Taper/reduce over time per protocol	Trial protocol; details on exact timing limited
Narrative/consensus guidance [34]	Steroids + IL-2RA induction (basiliximab)	Tacrolimus + MMF, then switch to mTOR or CNI + mTOR combination	Recommended to minimize CNI exposure in oncologic LT cases

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
