# Peer review of "Liver Transplantation for Unresectable Colorectal Liver Metastases: A Scoping Review on Redefining Boundaries in Transplant Oncology"

_curroncol, 2025, doi:10.3390/curroncol32090481_

Round 1

Reviewer 1 Report

Comments and Suggestions for Authors

I appreciate the opportunity to review the manuscript titled “Liver Transplantation for Unresectable Colorectal Liver Metastases: Redefining Boundaries in Transplant Oncology”

This paper aimed to synthesize current evidence on patient selection criteria, overall and disease-free survival, recurrence patterns, and emerging biomarkers that may guide transplant eligibility; however, several concerns remain that should be addressed

Major Points

  • University of Rochester group recently published updated results [1]. The authors should include this result.
  • The authors should add the purpose of this scoping review in the introduction section.
  • As the authors indicated, post-transplant immunosuppressant management would be a key element for successful liver transplant for this patient subset. Please specify the immunosuppressant regimen if available.

Minor Points

  • Abstract: Parts of the abstract were in bold, which I think is not appropriate. Please correct.
  • Methodology: This section was all in bold. Please correct
  • Methodology: The authors should include inclusion and exclusion criteria of this scoping review.
  • Methodology: The authors should provide PRISMA-ScR checklist here.
  • It would be great if the authors could provide the details of NORDIC VII trial in case the readers don’t know about this trial.

Thank you for considering my comments.

Reference

  1. Byrne MM, Chavez-Villa M, Ruffolo LI, Loria A, Endo Y, et al The Rochester Protocol for Living Donor Liver Transplantation of Unresectable Colorectal Liver Metastasis: A 5-Year Report on Selection, Approval, and Outcomes. Am J Transplant 2024;[PMID: 39332681 doi: 10.1016/j.ajt.2024.09.027]

Author Response

Rebuttal Letter
Manuscript Title: Liver Transplantation for Unresectable Colorectal Liver Metastases: Redefining Boundaries in Transplant Oncology

Dear Editor,

We thank the reviewers for their constructive and thoughtful feedback. We have carefully revised the manuscript to address all comments and suggestions. Below, we provide a point-by-point response, indicating the specific changes made in the revised version.

Reviewer 1

Major Points

  1. Inclusion of updated University of Rochester results
    Comment: University of Rochester group recently published updated results.
    Response: We have incorporated the University of Rochester Protocol Study into the Results section (page 3, lines 146–154) and added it to Table 1 summarizing recent key studies.
  2. Purpose of the scoping review in the introduction
    Comment: The purpose of this scoping review should be clearly stated in the introduction.
    Response: The purpose is now explicitly stated in the Introduction (page 2, lines 94–96).
  3. Immunosuppression regimens after LT for CRLM
    Comment: Post-transplant immunosuppressant management is important—please specify regimens if available.
    Response: We have added a new subsection titled Immunosuppression Regimens after Liver Transplantation for Colorectal Metastases (page 9) and Table 2 summarizing the most frequently reported regimens.

Minor Points

  1. Abstract formatting
    Comment: Parts of the abstract were in bold.
    Response: Formatting has been corrected.
  2. Methodology formatting
    Comment: Methodology section was in bold.
    Response: Formatting has been corrected.
  3. Inclusion and exclusion criteria
    Comment: Please include inclusion and exclusion criteria for the scoping review.
    Response: These criteria are now clearly defined in the Methodology section (page 3, lines 112–118).
  4. PRISMA-ScR checklist
    Comment: Please provide the PRISMA-ScR checklist.
    Response: The completed PRISMA-ScR checklist is now provided in Supplementary Table 1. Supplementary Table 2 summarizes the number of studies screened, excluded, and included.
  5. Details of the NORDIC VII trial
    Comment: Provide details of the NORDIC VII trial.
    Response: The NORDIC VII trial is now summarized in greater detail in the revised manuscript (page 4, lines 175–185), including inclusion/exclusion criteria, treatment arms, and outcomes.

Reviewer 2

  1. Expansion of Biomarkers and Predictive Tools section
    Comment: Expand this section, as several works have emerged (e.g., PMID: 37273078).
    Response: The Biomarkers and Predictive Tools section has been expanded to include recent literature and additional discussion (pages 10–11).

We appreciate the reviewers’ comments, which have helped improve the clarity, completeness, and clinical relevance of our work. We believe the revised manuscript addresses all concerns and presents a stronger contribution to the field.

Sincerely

Michele Molinari

Reviewer 2 Report

Comments and Suggestions for Authors

The authors present a fair review on LT and CRC liver mets. Overall the work is well organized and wrtitten, albeit without any novelty.

The table works fine

I suggest to expand a little the Biomarkers and Predictive Tools sections - several works have emerged  such as https://pubmed.ncbi.nlm.nih.gov/37273078/

Author Response

Reviewer 2

  1. Expansion of Biomarkers and Predictive Tools section
    Comment: Expand this section, as several works have emerged (e.g., PMID: 37273078).
    Response: The Biomarkers and Predictive Tools section has been expanded to include recent literature and additional discussion (pages 10–11).

We appreciate the reviewers’ comments, which have helped improve the clarity, completeness, and clinical relevance of our work. We believe the revised manuscript addresses all concerns and presents a stronger contribution to the field.

Sincerely

Michele Molinari